# Road Surface Classification Using a Deep Ensemble Network with Sensor Feature Selection

**DOI:** 10.3390/s18124342

**Published:** 2018-12-09

**Authors:** Jongwon Park, Kyushik Min, Hayoung Kim, Woosung Lee, Gaehwan Cho, Kunsoo Huh

**Affiliations:** 1Department of Automotive Engineering, Hanyang University, Seoul 04763, Korea; pjw2091@hanyang.ac.kr (J.P.); mks0813@hanyang.ac.kr (K.M.); hayoung.kim@hanyang.ac.kr (H.K.); 2Chassis System Control Development Team, Hyundai Motor Company, Gyeonggi-do 18280, Korea; woosung.lee@hyundai.com; 3Autonomous Vehicle Technology Laboratory, SW Part, CTO, LG Electronics, Seoul 07796, Korea; gaehwan.cho@lge.com

**Keywords:** road classification, ensemble learning, recurrent neural network, feature selection

## Abstract

Deep learning is a fast-growing field of research, in particular, for autonomous application. In this study, a deep learning network based on various sensor data is proposed for identifying the roads where the vehicle is driving. Long-Short Term Memory (LSTM) unit and ensemble learning are utilized for network design and a feature selection technique is applied such that unnecessary sensor data could be excluded without a loss of performance. Real vehicle experiments were carried out for the learning and verification of the proposed deep learning structure. The classification performance was verified through four different test roads. The proposed network shows the classification accuracy of 94.6% in the test data.

## 1. Introduction

With the recent development of vehicle control technology, the number of vehicles with active suspension is increasing. The active suspension system changes the characteristics of the suspension depending on the driving condition to secure the ride comfort and steering stability. Most active suspension systems apply low damping characteristics for comfort on bumpy roads and high damping characteristics for steerability on flat roads. Thus, it is very important for active suspension control to obtain the information of road shape such as bump, pothole, etc.

Recently, deep learning is being widely studied in the various fields such as image processing, natural language processing and reinforcement learning [1]. Deep learning has also been utilized for identifying road conditions recently. Deep learning is a learning-based method using a neural network structure with multi hidden layers. Many processing units, which is called Perceptron, are connected each other and each perceptron consists of nonlinear function. A large number of perceptron operations can effectively approximate complex nonlinear functions, resulting in high performance improvements in various fields. Moreover, it is showing good performance in sequential data processing, so deep learning is actively applied to sensor data analysis recently.

In this study, an algorithm, which estimates the type of the road surface by a deep learning technique, is proposed. Four different types of the roads (flat road, sinusoidal road, manhole and pothole, bump) are considered for this study. Various sensor data is utilized for training and verification, which are available in the Controller Area Network (CAN) of a real vehicle.

The main contributions of this paper can be summarized as follows:Deep ensemble structure with LSTM is designed for the road surface classification. By using the Recurrent Neural Network (RNN) structure with LSTM, sequential sensor data are processed and a classification result is generated. The ensemble structure is used to avoid overconfidence estimation and overfitting.Network input is constructed with only the sensor data that can be obtained from the in-vehicle network. The proposed algorithm can be applied to vehicles with active suspension without additional sensors.The feature selection technique is applied to determine the importance of each piece of sensor data. Sensor data are selected as an input depending on its importance, and the number of inputs can be reduced without a loss of performance.

The remainder of this paper is organized as follows: Section 2 reviews the literature of road surface classification. Section 3 describes the methodology used in this study. Section 4 explains the structure and learning method of the proposed network. Section 5 explains the verification process through experiments and discussion. Conclusions and future work are given in Section 6.

## 2. Related Work

The research to determine the road type has mostly been based on the dynamics of the vehicle suspension and chassis. A lot of studies have been carried out to estimate the road profile mainly through the suspension model. Imine et al. [2] design a sliding mode observer to estimate the road profile where the height of the road surface is measured by an additional profiler and is compared with the estimated value. The road profile is well estimated in a section where the elevation of the road is small, but the performance of the observer is insufficient in the section where the elevation of the road is large. Doumiati et al. [3] obtain the road profile from the profiler and it shows good performance, but this technique is limited to only large commercial vehicles.

Instead of estimating the road profile directly, there is an approach to classify the road profile based on the Power Spectral Density (PSD) of the road. Qin et al. [4] estimate a road profile through Kalman filter and wavelet transform to perform the classification. Wang et al. [5] estimated the road level in various road conditions by using adaptive Kalman filter considering uncertainty of the road condition. There are also studies to improve classification performance by installing additional sensors. Kumar et al. [6] use an additional LIDAR sensor that measures the surface of the road to classify the type of road surface.

In recent years, many studies are being conducted to classify road surface through machine learning. Mou et al. [7] proposed a method to classify five different types of the road surface (grass, asphalt, gravel, pavement and indoor) using a learning-based classifier and controls speed of the vehicle with respect to the classification result. The result of classification is derived by combining the results of multiple classifiers based on different sensors of vibration sensor, vision sensor and laser sensor to classify. Support Vector Machine (SVM) and Gaussian Mixture Models (GMMs) are used to classify the type of the terrain rather than the type of the road surface for caterpillar robot. Qin et al. [8] performed road surface classification for a semi-active suspension system using a deep neural network structure which consists of a sparse autoencoder and softmax classifier. Based on the Carsim simulation model, data is collected to classify six different road levels defined by ISO (International Organization for Standardization). Yousefzadeh et al. [9] estimated the road profile using an artificial neural network whose architecture consists of three linear hidden layers. Seven different kinds of acceleration (roll, bounce, pitch and four wheels) are used as input and data is collected from the vehicle model of ADAMS software (MSC Software Corporation, Newport Beach, CA, USA). Solhmirzaei et al. [10] conducted road profile estimation using a wavelet neural network that uses wavelet basis function as an activation function instead of sigmoid function. Seven vehicle accelerations and several state variables from the previous time steps are obtained by a 7-DOF (Degrees of Freedom) vehicle dynamics model using MATLAB (MathWorks, Natick, MA, USA) and used as inputs for the network. The output is the road profile and training data is composed of four different kinds of road surfaces using an ISO8608 standard.

## 3. Methodology

### 3.1. Recurrent Neural Network (RNN)

Among deep learning algorithms, Recurrent Neural Network (RNN) is a specialized model for analyzing data that is related with time, such as sequential data. RNN model transfers its output to the next time step by multiplying with a weight. However, this structure causes a vanishing gradient problem and it was difficult to learn long sequences. Models such as Long-short Term Memory (LSTM) and Gated Recurrent Units (GRU) are used to solve this problem [11,12,13]. In this study, vehicle sensor data is used as the input of the deep learning model. Sensor data is also correlated with time. Therefore, an RNN with an LSTM cell is used as the network model of this study.

### 3.2. Ensemble Learning

In general, the deep learning algorithm has a problem of overconfidence on the output and an overfitting problem that adversely affects the generalization because it is too closely fitted to the training data. These problems decrease the performance of the deep learning algorithm and make it hard to estimate robustly. Ensemble, which is mainly used in machine learning, is a technique for deriving the results from more than one model and synthesizing the results of each model to determine the final result [14,15,16]. This is a technique that alleviates the overconfidence and overfitting problems. In this study, ensemble structure, which consists of the same RNN networks, is used as a network model. Figure 1 shows the concept of ensemble structure.

## 4. Deep Learning for Road Surface Classification

### 4.1. Network Input

In this study, data is obtained by using an IMU sensor, steering angle sensor, vehicle speed sensor and acceleration sensor. Fourteen pieces of sensor data were initially used as input as follows:IMU: Longitudinal acceleration, Lateral acceleration, Yaw rate;Steering sensor: Steering angle, Steering angular velocity;Speed sensors: Speed of each wheel;Suspension sensors: Vertical acceleration on the front left and front right wheel, Vertical acceleration of sprung mass on the front left, front right and rear left.

Deep neural network has the ability to analyze unknown features and use them to solve the problems. In this study, many available in-vehicle sensor data is used to make the input data. It is expected that the deep neural network is able to find the suitable features among those sensor data.

Sensor data of the vehicle have various units, so the range of each value is different. Using them as a network input is likely to have an adverse effect on network training and performance. To overcome this, all the network inputs are normalized to [–0.5, 0.5]. The equation of the normalization is as follows:(1)$$s_{i,normalized}=\frac{1}{max_{i}-min_{i}}\left(s_{i}-\frac{1}{2}\left(max_{i}-min_{i}\right)\right),$$
where *i* is index of each sensor, max, min are maximum and minimum value of each sensor data, respectively. *s* is data of each sensor.

Sensor vector x is defined as $x=\left[s_{1},\ldots ,s_{N_{s}}\right]$ and $N_{s}$ is the number of sensors. Sensor vectors are stacked in time sequence and used as the input of RNN network. Finally, the network input is the matrix of size $N_{s}\times L_{s}$ and $L_{s}$ is the length of the input sequence:(2)$$\mathrm{Input}=\left[x_{1};\ldots ;x_{L_{s}}\right].$$

### 4.2. Network Structure

The structure of the LSTM cell which is used in this study is illustrated in Figure 2. The LSTM network recursively operates according to the sequence, and the cell state transfer the information through the sequence. The input data of the LSTM cell consists of the cell state, output of the previous step and input data of the current step. There are three gates inside LSTM and the gates manage the cell state information. The first gate is called the forget gate, and the forget gate selects the part of cell state data to be removed. The second gate is the input gate and it determines the data to be added to the cell state. Lastly, the output gate generates output data using the cell state. This process is computed sequentially through the following equations [13]:(3)$$\begin{array}{c} f_{t}\;=\;\sigma \left(W_{f}\cdot \left[h_{t-1},x_{t}\right]+b_{f}\right),\\ i_{t}\;=\;\sigma \left(W_{i}\cdot \left[h_{t-1},x_{t}\right]+b_{i}\right),\\ C_{i,t}\;=\;\tanh\left(W_{C}\cdot \left[h_{t-1},x_{t}\right]+b_{C}\right),\\ C_{t}\;=\;f_{t}*C_{t-1}+i_{t}*C_{i,t},\\ o_{t}\;=\;\sigma \left(W_{o}\cdot \left[h_{t-1},x_{t}\right]+b_{o}\right),\\ h_{t}\;=\;o_{t}*\tanh\left(C_{t}\right), \end{array}$$
where *t* is the time step in input sequence. x,h and *C* are the input, output and cell state of the LSTM cell, respectively. *f* is the forget selection vector. *i* is the input selection vector. $W_{f},W_{i},W_{c},W_{o}$ are the weight variables in forget, input and output gate. $b_{f},b_{i},b_{c},b_{o}$ are the bias variables in forget, input and output gate. $\sigma \left(\cdot \right)$ denotes the sigmoid function. $\tanh\left(\cdot \right)$ denotes the hyperbolic tangent function. ∗ denotes the element-wise multiplication.

In this study, a multi-layer LSTM network is designed by stacking two LSTM cells and two dropout layers. The dropout layer is applied to reduce the overfitting problem [17,18]. Figure 3 shows the structure of the proposed network. Each LSTM cell has a cell state size of 1024. At the output layer, the softmax function is applied to calculate the probability for each class. The output of LSTM network is calculated by the following equation:(4)$$h_{s}=\;W_{s}\cdot h_{2,L_{s}}+b_{s},$$
(5)$$p=\mathrm{softmax}\left(h_{s}\right)=\;\frac{e^{h_{s}}}{\sum_{j=1}^{N_{c}}e^{h_{s,j}}},$$
where *p* is the output of the multi-layer LSTM network and $N_{c}$ is the number of the output class. Subscript *j* means the index of each class. $W_{s}$ and $b_{s}$ are the weight and bias variable in softmax layer, respectively.

Several multi-layer LSTM networks were stacked for ensemble learning. An ensemble network consists of several single networks and each single network has the structure of Figure 3. The output of ensemble network is the average of the outputs of each single network as shown in the following Equation:(6)$$p_{e}=\frac{1}{N_{e}}\left(\sum_{z=1}^{N_{e}}p_{z}\right),$$
where $p_{e}$ is the output of ensemble network and $N_{e}$ is the number of single network. Subscript *z* means the index of each single network.

### 4.3. Network Training

Training of the ensemble network is performed by the training dataset. Each single network in ensemble is trained through a different batch set which is made by random sampling in the training dataset. It prevents each network from being trained as the same variables. After ensemble learning, each network has the same structure and the same classification task but has different variables. By combining the results from these networks, a more generalized model can be obtained for the classification task. Each network is trained using softmax cross entropy and the equation of loss function is as follows:(7)$$L_{z}=-\sum_{j=1}^{N_{c}}y_{j}\;log\left(p_{z,j}\right).$$

Training is performed by an Adam optimizer [19] to minimize the loss in Equation (Equation 7). Each network has a different value of $L_{z}$. Hyperparameters, which are used for training, are listed in Table 1. The entire learning process of the ensemble network is shown in Algorithm 1 and Figure 4. The word ’variables’ in Figure 4 indicates the weights and bias in each network.

**Algorithm 1** Training Process of the Ensemble Network.
1:Establish the ensemble network through the single networks (multi-layer LSTM network)2:Initialize the weights of each single network in ensemble network3:**for** iterations **do**4:  **for**
z=1 to $N_{e}$
**do**5:    Make a batch set $b_{z}$ by random sampling in training data6:    Only the variables in single network *z* are assigned as trainable variables7:    Minimize the loss $L_{z}$ of the batch set $b_{z}$ through the Adam optimizer8:  **end for**9:
**end for**
10:Verify the performance with test data


### 4.4. Feature Selection

The network is trained by 14 sensor data available in the vehicle as input. However, among these data, several variables are not needed for road surface classification and they increase computation and degrade the performance. Feature selection is applied to select important features among the input data. It can simplify the model and accelerate training of the network. In this study, each sensor data is used as a feature and find out which sensor data is important for the performance of the road surface classification. For the feature selection, importance weight [20], which represents importance of each sensor data, is expressed as the weight variables and they are added to the input layer with the range of [0,1]. Each weight variable, $w_{i}$, is scalar and corresponds one to one with each piece of sensor data $\left(i=1,\ldots ,N_{s}\right)$. This weight variable, $w_{i}$, is multiplied with the sequence data from the sensor and the weighted sensor vector, $x_{w}$, is defined as follows.
(8)$$x_{w}=\left[w_{1}s_{1},\ldots ,{w_{N}}_{s}{s_{N}}_{s}\right].$$

Using this weighted sensor vector, weighted input matrix is calculated as follows:(9)$$\mathrm{Input}=\left[x_{w,1};\ldots ;x_{w,L_{s}}\right].$$

Figure 5 shows the network architecture including the importance weight layer.

The training of the importance weight layer is done separately from the training of the ensemble network. If the ensemble weight and importance weight are trained at the same time, weights of ensemble network intervene in the estimation of the result and it is difficult to efficiently learn the importance weight. For this reason, weights of the ensemble network in Section 4.3 is used without any change and only the importance weights are trained. The loss function for training importance weight is as follows:(10)$$L_{fs}=\;-\sum_{j=1}^{N_{c}}y_{j}\;log\left(p_{j}\right)+\lambda \sum_{j=1}^{N_{c}}w_{i}.$$

The loss function includes the sum of importance weight in addition to the softmax cross entropy. In this way, the network maintains the accuracy of the classification and reduces the importance weight of unnecessary data. λ is set to 0.01 as a hyperparameter. After the training, if the importance weight of a data is close to 0, it means that the data is less significant to the network performance. On the contrary, the data is important for the performance of the network, if its importance weight is close to 1.

## 5. Experimental Results and Discussion

In this section, the experiment procedure and the training results are described. Results of this study include a single network result, ensemble network result and feature selection result. In addition, the process of determining the length of sequence used in the LSTM network is included.

### 5.1. Experiment

In this study, data were acquired from four different types of test roads for training and testing of the deep learning algorithm. The data was obtained from the actual vehicle experiments on each road. The description of the test roads is illustrated in Figure 6.

The overall structure of the experiment procedure is shown in Figure 8. The experiment was conducted at various speeds and the maximum speed was limited for safety on some roads. Input data is obtained at every 10 ms using CAN. In order to verify the performance of the deep learning algorithm, test data was acquired through additional experiments. When the test data was obtained, the vehicle speed was different from the training data to verify the overfitting of the network. Specific information of the experiment is listed in Table 2. The examples of sensor data from experiment are shown in the Appendix A.

Labeling was performed on the experimental data and is determined based on the position of the front wheel. The class has an integer value from 0 to 3, and each class corresponds to the road where the actual vehicle experiment was conducted:0: Flat road,1: Sinusoidal road,2: Manhole and pothole,3: Bump.

### 5.2. Experimental Result

In this section, training results of the proposed algorithm and verification are described. In order to verify the effectiveness of the ensemble learning, performance of the ensemble network is compared with a single network. The single network is the single LSTM network in Figure 3. Length of the sequence is 80 for both single network and ensemble network. The result of comparison is shown in Table 3. Training accuracy of both networks shows over 99% and it means that both networks are well trained. However, test accuracy shows the difference in performance between the two networks. The ensemble network shows classification accuracy about 4% better than the single network and demonstrates that overfitting decreases when applying ensemble learning. Performance of the two networks is compared through the confusion matrix. The confusion matrix represents the classification result for the test data. Figure 9 and Figure 10 show that all the classification accuracy improves through the ensemble learning, in particular, the case of the sinusoidal road is improved significantly.

The length of the input sequence, which is applied to the LSTM, is a factor directly affecting the performance of the network. The input sequence is the data that accumulates sensor data every 0.01 s, and the sequence length determines how many seconds of data are used in the network. Generally, performance of the network is better when longer sequence length is applied because the result is classified with more data. However, this trend can be saturated in a certain length because old data may not be useful for classification. In addition, sequence length directly affects the computation time of the network. The longer the sequence length, the larger the size of the input, which increases the computation time. It is important to set the sequence length in consideration of such advantages and disadvantages. In this study, various lengths of the sequence are tested to find the proper sequence length and this result is shown in Figure 11. Test accuracy shows that the performance improves with increasing of the sequence length in the range of 10 to 80. However, this tendency is diminished after 80. The difference between the train accuracy and the test accuracy is reduced as the sequence length is increased. This shows that the overfitting decreases and performance improves as the sequence length increases. In this study, the sequence length is set to 80 to ensure high performance of the network and generalization of the model.

The importance of weight training results for feature selection are shown in Table 4. The importance weight value indicates the importance of each sensor data. The lateral acceleration sensor, speed sensor, yaw rate sensor and vertical acceleration sensors of the sprung mass show high importance. On the other hand, the vertical acceleration sensors of the unsprung mass, steering angle speed sensor and longitudinal acceleration sensor show low importance in classification. In this study, sensor data with an importance weight over 0.4 are selected as the input features. After the feature selection, the input data of the sensors are reduced from 14 to 10.
Lateral acceleration, yaw rate, steering angle, speed of each wheel, vertical acceleration at front left, front right, rear left of the sprung mass.

With the new input of dimension 80 × 10, the classification accuracy of the network trained through the new features is shown in Figure 12. Total classification accuracy in test data is 94.6%. Although the number of inputs is reduced, the network performance remains robust. This means that the unnecessary data in the classification are successfully removed through the feature selection technique.

The test data were sequentially predicted to identify the classification results continuously over time. The network with feature selection and ensemble learning is used. Figure 13, Figure 14, Figure 15, Figure 16, Figure 17, Figure 18 and Figure 19 show some of the test results. In the graphs, cyan lines represent true label of the class and black dots represent the prediction result of the network. The numbers in the upper left corner indicate the classification accuracy of each class in that experiment. The results also show that higher speed induces higher accuracy. Higher vehicle speed causes large difference in sensor data, which makes it easier to be distinguished by the network.

## 6. Conclusions

In this study, a deep learning method based on sensor data is proposed to identify road surface for vehicles. LSTM is applied to process sequential sensor data and ensemble learning is applied for robust estimation. In addition, feature selection is used to determine the importance among the sensor data, and unnecessary sensor data is excluded from the input of the network. In order to verify the proposed algorithm, the experiment was carried out on four different test roads. Train data and test data were obtained in separate experiments to confirm the generalization of the model. Test data was used for verification and it shows that the ensemble learning improves classification performance and reduces the overfitting problem. The effectiveness of the feature selection is demonstrated with the classification result that the performance is maintained with less sensor data. The network with ensemble learning and feature selection shows 94.6% classification accuracy on the road surface.

The future work of this study can be considered in two ways: first, the sensor data was acquired in four kinds of test roads for the proposed algorithm. Therefore, the dataset is limited in those test roads. However, there are many kinds of road conditions in the actual road environment. For this reason, the proposed algorithm can be extended by applying more diverse road conditions. Second, after the road surface classification through the proposed algorithm, the vehicle can be controlled according to the estimated road condition. Therefore, the suspension control strategy to enhance stability and steerability has to be considered as a future work.

## Figures and Tables

**Figure 1 sensors-18-04342-f001:**
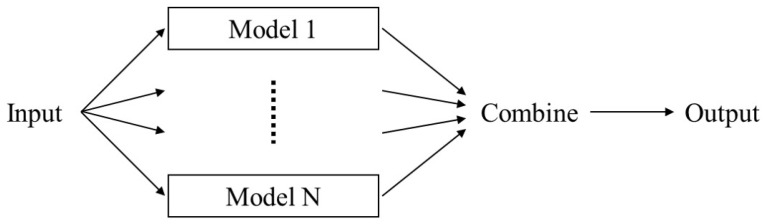
Conceptual diagram of ensemble structure.

**Figure 2 sensors-18-04342-f002:**
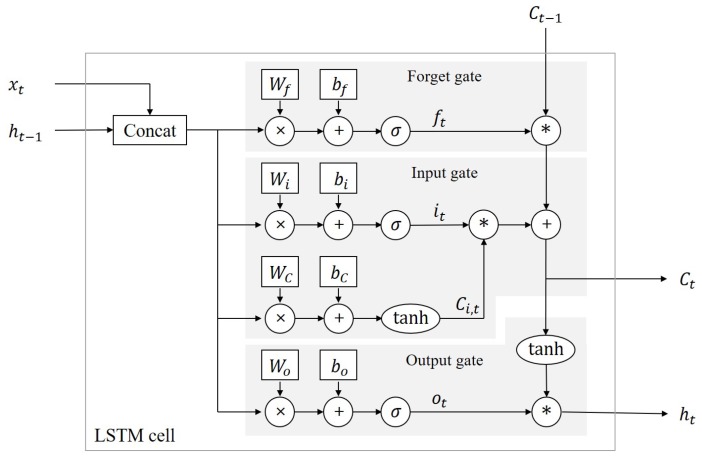
Structure of the LSTM cell.

**Figure 3 sensors-18-04342-f003:**
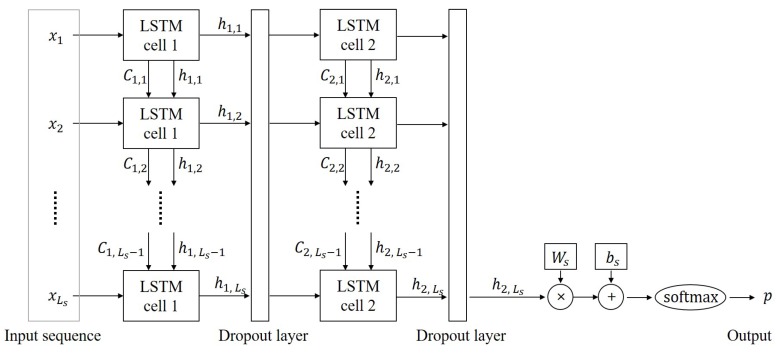
Structure of the multi-layer LSTM network.

**Figure 4 sensors-18-04342-f004:**
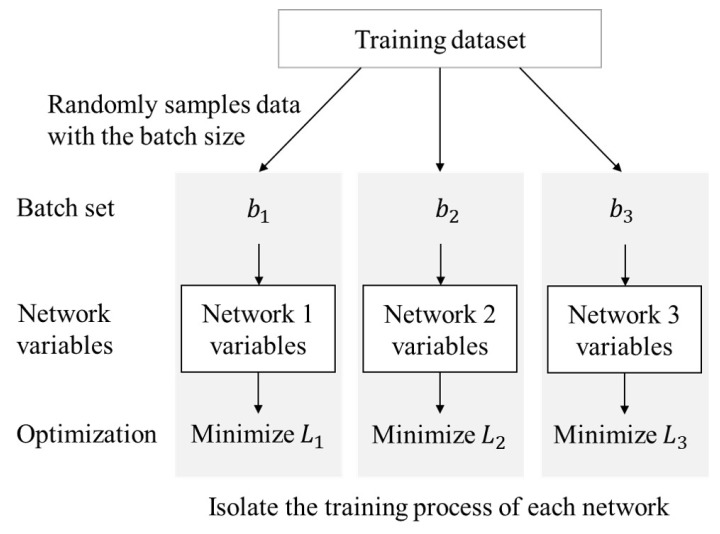
Multiple network learning process for the ensemble network.

**Figure 5 sensors-18-04342-f005:**
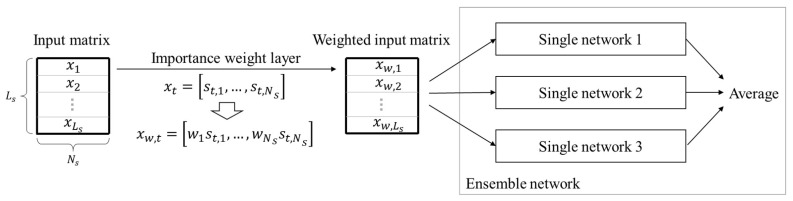
Structure of the ensemble network with importance weight layer.

**Figure 6 sensors-18-04342-f006:**
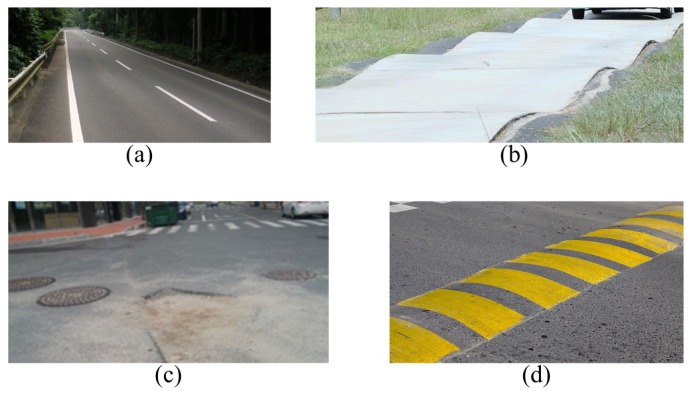
(**a**) flat road: flat, asphalt road; (**b**) sinusoidal road: sine wave road (amplitude: 0.05 m, wavelength: 10 m); (**c**) manhole and pothole: road with many manholes and potholes; (**d**) bump: bumps on the flat road. Three different types of bumps are used as shown in Figure 7.

**Figure 7 sensors-18-04342-f007:**
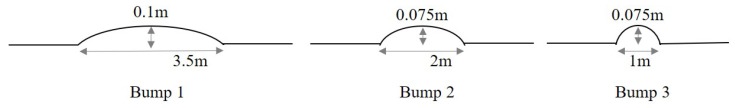
Shape of bumps.

**Figure 8 sensors-18-04342-f008:**
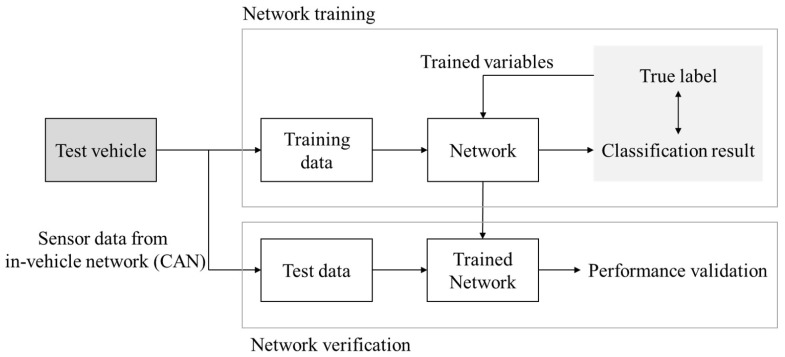
Experiment and learning process.

**Figure 9 sensors-18-04342-f009:**
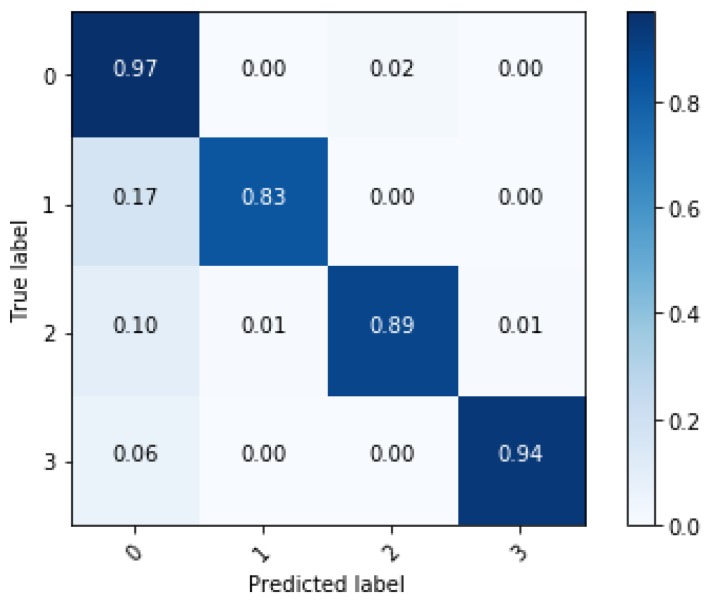
Confusion matrix of classification results from the single network.

**Figure 10 sensors-18-04342-f010:**
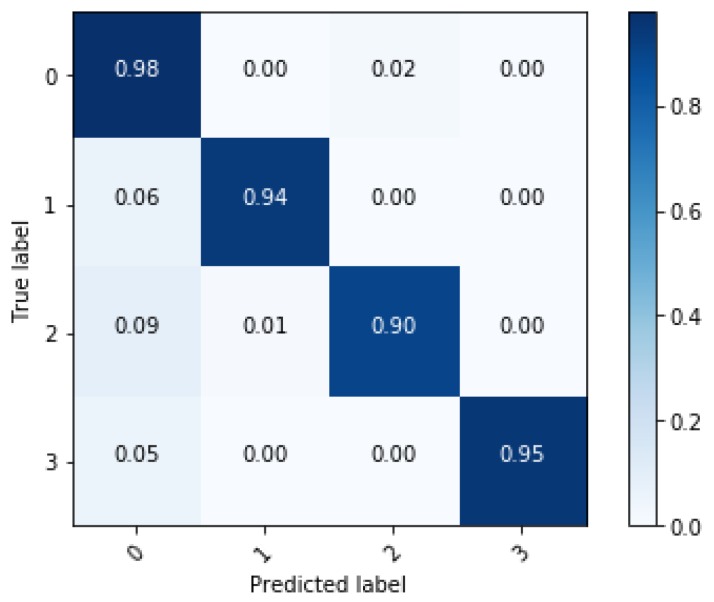
Confusion matrix of classification results from the ensemble network.

**Figure 11 sensors-18-04342-f011:**
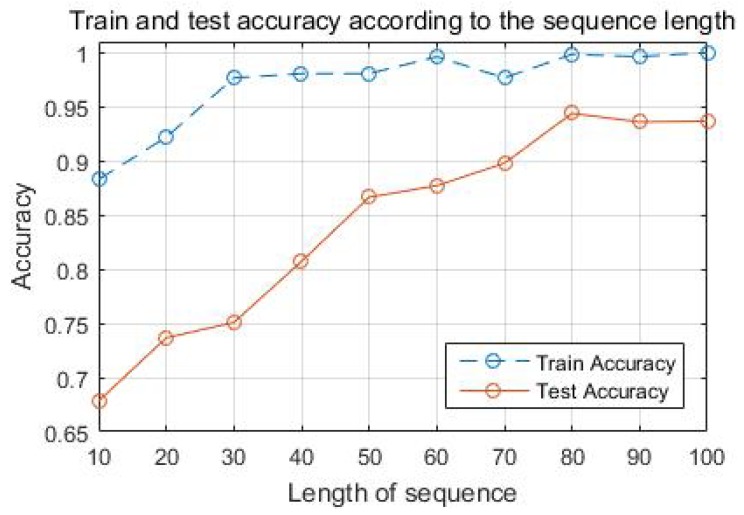
Classification accuracy with the sequence length.

**Figure 12 sensors-18-04342-f012:**
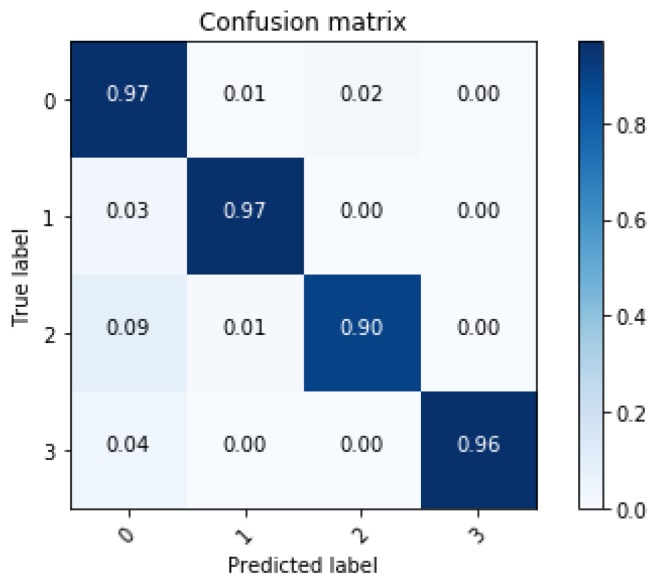
Confusion matrix after feature selection.

**Figure 13 sensors-18-04342-f013:**
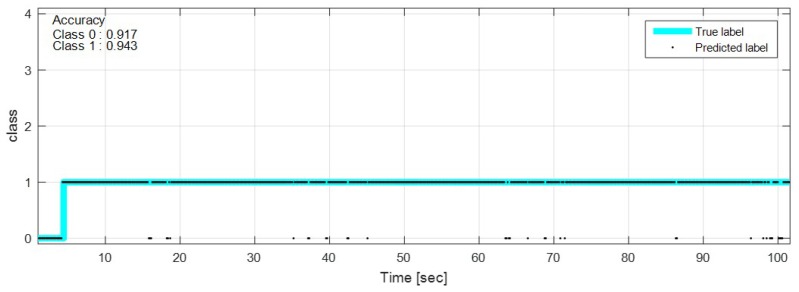
Prediction result on sinusoidal road, 20 km/h.

**Figure 14 sensors-18-04342-f014:**
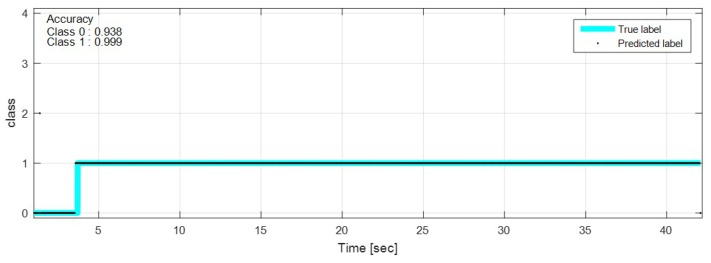
Prediction result on sinusoidal road, 50 km/h.

**Figure 15 sensors-18-04342-f015:**
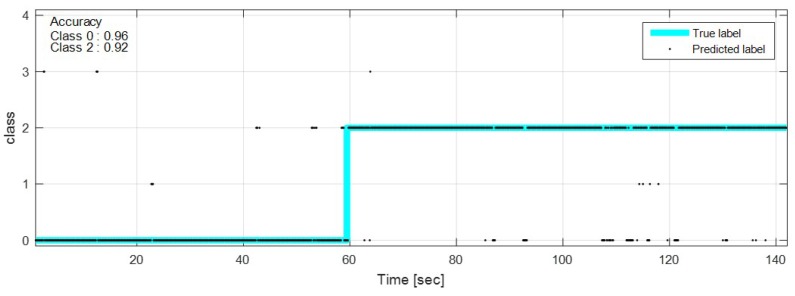
Prediction result on manhole and pothole road, 20 km/h.

**Figure 16 sensors-18-04342-f016:**
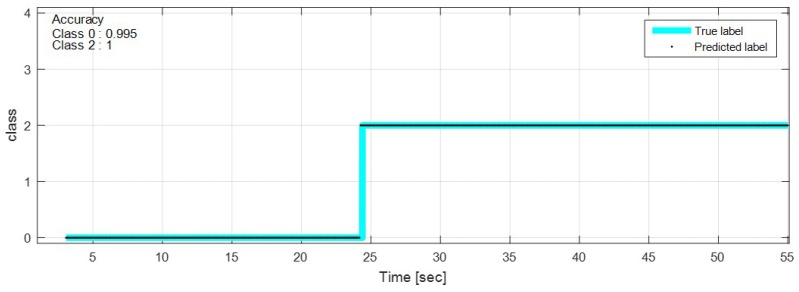
Prediction result on manhole and pothole road, 50 km/h.

**Figure 17 sensors-18-04342-f017:**
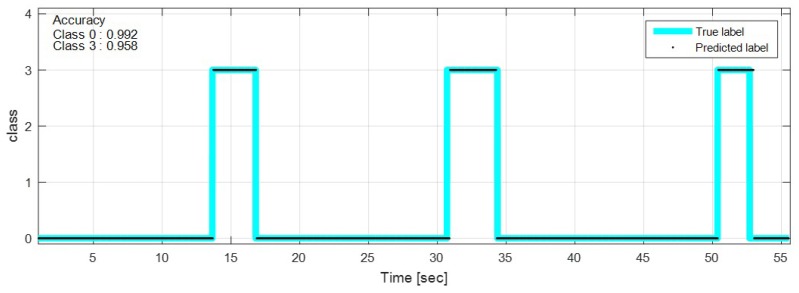
Prediction result on bump road, 10 km/h.

**Figure 18 sensors-18-04342-f018:**
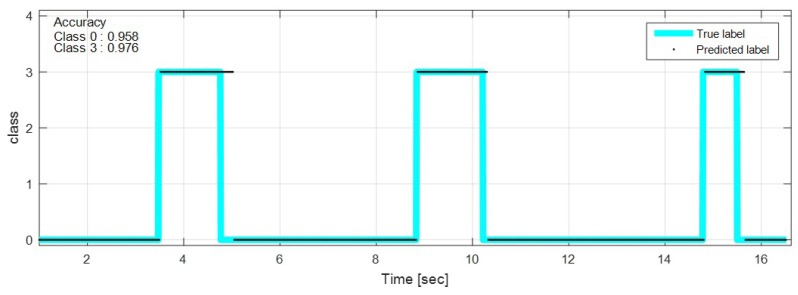
Prediction result on bump road, 35 km/h.

**Figure 19 sensors-18-04342-f019:**
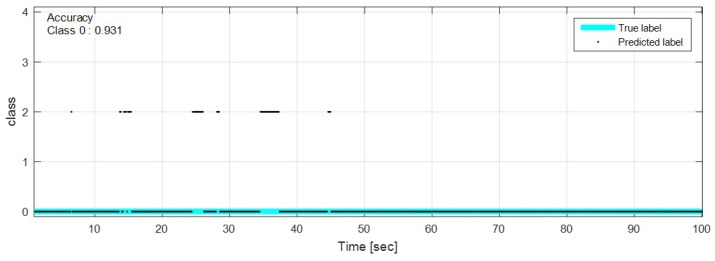
Prediction result on flat road, free speed (0∼100 km/h).

**Table 1 sensors-18-04342-t001:** Hyperparameters used in the training process.

Name	Value
Number of single networks, $N_{e}$	3
Keep probability of dropout layers	0.33
Number of iterations	20,000
Batch size	256
Learning rate	0.0001

**Table 2 sensors-18-04342-t002:** Experimental scenario.

	Types of Road	Vehicle Speed for Data Acquisition (km/h)	Total Number of Data
Training data	Flat road	20, 40, 60, 80	380,556 steps (63.4 min)
	Sinusoidal road	20, 40, 60	
	Manhole and Pothole	20, 40, 50	
	Bump	10, 20, 30	
Test data	Flat road	0∼100	129,873 steps (21.6 min)
	Sinusoidal road	20, 30, 40, 50, 60	
	Manhole and Pothole	15, 20, 30, 40, 45, 50	
	Bump	10, 20, 25, 30, 35	

**Table 3 sensors-18-04342-t003:** Classification accuracy of the single network and the ensemble network.

	Train Accuracy	Test Accuracy
Single network	99.6%	90.6%
Ensemble network	99.8%	94.4%

**Table 4 sensors-18-04342-t004:** Classification accuracy of the single network and the ensemble network.

Sensors	Location	Importance Weight
Vertical acceleration of sprung mass	Front left	0.9514
	Front right	0.9999
	Rear right	0.9065
Vertical acceleration of wheels	Front left	0.2169
	Front right	1.558 × 10${}^{-5}$
Longitudinal acceleration	-	3.826 × 10${}^{-6}$
Lateral acceleration	-	0.9746
Yaw rate	-	0.9988
Steering angle	-	0.4833
Steering angular velocity	-	2.105 × 10${}^{-4}$
Wheel speed	Front left	0.8626
	Front right	0.8892
	Rear left	0.8347
	Rear right	0.5921

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
