# Peer review of "Road Surface Classification Using a Deep Ensemble Network with Sensor Feature Selection"

_sensors, 2018, doi:10.3390/s18124342_

Reviewer 1 Report

  1. Tab. 4 - Importance weight for "Vertical acceleration of front right wheel" is equal to 0.0000.  Please explain this as it looks strange, or correct if the value is wrong.

  2. More information about design of experimental verification is necessary (photos of the road surfaces, information about test vehicle, etc).

  3. While theoretical part is rather broad, the experimental part needs some enhancement.

Author Response

The authors are grateful for the kind review. We believe that the manuscript was improved based on your suggestions. The answer to the review can be found in the PDF file. Again, thank you very much for your comments.

Reviewer 2 Report

The text is in general well written. Some minor mistakes are found, but very minor and can be fixed with a careful review. I liked the paper and I have just some minor comments to improve it before the final acceptance.

  1. Please, refrain from using the first person, “we, our, us”, please prefer using third person or passive voice instead.

  2. Please, state clearly the contributions for the paper in the introductory section.

  3. I think Section 4.3 is a bit superficial. You could elaborate more in explaining the training phase.

  4. In order to ensure reproducibility, I would ask the authors to elaborate more in the explanation of the experimental setup in Section 5.1. Please, provide more details so that other researchers can easily reproduce your experiments.

  5. Directions for future work are missing in the conclusion. Please include at least a sentence about it. 

Author Response

(The authors gave the same response as above.)
